# Factors Driving Individuals’ Attitudes toward Sugar and Sweet-Tasting Foods: An Analysis within the Scope of Theory of Planned Behavior

**DOI:** 10.3390/foods13193109

**Published:** 2024-09-28

**Authors:** Hatice Pınar Kural Enç, Meryem Kahrıman, Cansu Gençalp, Salim Yılmaz, Gizem Köse, Murat Baş

**Affiliations:** 1Department of Nutrition and Dietetics, Institute of Health Sciences, Acibadem Mehmet Ali Aydinlar University, Istanbul 34752, Turkey; hatice.enc@live.acibadem.edu.tr; 2Department of Nutrition and Dietetics, Faculty of Health Sciences, Acibadem Mehmet Ali Aydinlar University, Istanbul 34752, Turkey; meryem.kahriman@acibadem.edu.tr (M.K.); cansu.gencalp@acibadem.edu.tr (C.G.); gizem.kose@acibadem.edu.tr (G.K.); 3Department of Healthcare Management, Faculty of Health Sciences, Acibadem Mehmet Ali Aydinlar University, Istanbul 34752, Turkey; salimyilmaz142@gmail.com

**Keywords:** theory of planned behavior, sugar intake, attitude, intention, sociodemographic factors

## Abstract

Public health authorities are undertaking initiatives aimed at reducing sugar intake because it is linked to detrimental health outcomes. Individuals’ attitudes and intentions toward sugar can be significant factors affecting intake. Therefore, we here aimed to evaluate individuals’ attitudes and intentions toward sugar intake by combining the theory of planned behavior with different frameworks. Accordingly, we developed an online questionnaire and reached 940 participants. We observed that subjective norms (β = −0.140; *p* < 0.001) and perceived behavioral control (β = −0.138; *p* < 0.001) as defined in the theory of planned behavior significantly affected attitude. Subsequently, examining the effect of attitude (β = −0.209; *p* = 0.018) and intention (β = 0.717; *p* < 0.001) on sugar intake showed that intention had a positive effect, whereas attitude had no significant effect. Mediation analysis showed that attitude influenced sugar intake through intention (β = −0.286; *p* = 0.001). Furthermore, we determined that components including habits, perceived understanding, perceived nonautonomy, negativity, and apathy can affect attitude toward sugar intake (*p* < 0.001). Analyzing the effects of sociodemographic factors revealed that sugar intake was lower in individuals with food allergies (*p* < 0.05). In conclusion, these findings suggest that the theory of planned behavior, when combined with different frameworks, can be effective in predicting sugar intake and individuals’ intentions.

## 1. Introduction

Dietary sugar is divided into the following different classes: total, free, and added. Total sugar refers to the sugar present in food, obtained by any means. Free sugars are sugars that occur naturally in fruits and vegetables. Moreover, the term includes added sugars that are added to foods by the producer or consumer [1]. Sugar provides palatability and reward, especially in terms of taste [2]. Owing to this feature, its consumption among individuals has become increasingly widespread [3]. Previous studies reported that added sugar intake ranges from 8% to 14% of energy and emphasized that this intake varies according to different factors, including age groups, race, physical activity level, and body mass index (BMI) [3,4]. Data from the Turkish Nutrition and Health Survey reported that 16.4% of individuals in the 10–20% range and 3.8% in the ≥20% category contribute sugar to their daily energy intake [5]. These data highlight that a significant proportion of the population consumes more sugar than the World Health Organization recommendation of <10% [6]. Studies in the literature have emphasized the relationship between increased free sugar intake and obesity, metabolic diseases, cardiovascular diseases, cancer, and oral and dental health problems [7,8,9,10]. Owing to these harmful effects, various authorities have issued guidelines and recommendations on reducing sugar intake [6,11]. Additionally, initiatives to reduce sugar intake have largely focused on reducing sugar availability in the food supply and on behavioral changes to steer consumers toward low sugar-containing foods [12].

Consumers have reported positive perceptions of sugar and sugar-sweetened foods and beverages, including the experience of pleasure [13], provision of energy [14], and some perceived health benefits [15]. Several studies have been conducted to identify modifiable cognitive and motivational constructs that predict sugar intake to assess whether these perceptions influence sugar intake and plan behavioral interventions for reducing sugar intake. These studies have revealed that cognitive and motivational factors, including habit, self-control, perceived behavioral control, personal impact, and personal management, influence sugar intake [14,16,17,18]. The theory of planned behavior is the framework used for predicting these behaviors of individuals [19]. This theory suggests that attitude, subjective norms, and perceived behavioral control can largely predict individuals’ intention to perform behaviors [19,20]. It states that the more positive and greater individuals’ attitudes, subjective norms, and perceived behavioral control regarding behavior are, the greater the intention will be [19]. The hypotheses of the theory of planned behavior have been evaluated in previous studies for various health behaviors, such as sugar intake [21] and dietary behaviors [22]. However, it is of note that the theory of planned behavior has some criticisms in the literature. One of them is that the effect of intention on behavior is not highly significant [23]. Another criticism is the idea that it predicts frequently performed behaviors, including dietary behavior and physical activity, with lower variance; therefore, it may not be very suitable for predicting such behaviors [24]. Therefore, applying the theory of planned behavior in combination with other theories and frameworks is essential.

Sociodemographic factors are one of the significant factors that are discussed as being influential on attitude toward sugar intake. In addition, studies underscore that individual factors, including age, sex, ethnicity, and education level are effective in attitude toward sugar intake [25]. Data emphasizing that these factors do not have a significant effect are also noted [16]. Considering these findings, the relationship between sociodemographic factors and sugar intake behavior remains unclear, and the role of these factors should be clarified. Therefore, we here aimed to evaluate the effects of variables (Table 1), including habits, subjective norms, perceived behavioral control, self-control, and personal impact on sugar intake, by considering the theory of planned behavior [19,20] and previous studies [14,16,17,26]. Furthermore, we aimed to examine the effects of sociodemographic factors, including age, sex, and education level. In this direction, we hypothesized that individuals’ sugar intake will be influenced by various factors, including subjective norms, perceived behavioral control defined in the theory of planned behavior, and sociodemographic factors.

## 2. Materials and Methods

### 2.1. Participants

Participants were recruited on social media, which was supplemented using the snowball sampling method. To calculate the sample size, a 5:1 subject-to-item ratio was used [27]. Accordingly, the minimum sample size, comprising 78 items, was 390 participants.

Overall, in this study, 940 individuals aged >18 years were included. Every participant signed an informed consent form and voluntarily consented to take part in the study. Data were collected online through a questionnaire (Appendix A) prepared on Google Forms. The study protocol was approved by Acibadem University Medical Research Ethics Committee (2024-11/484), and this study was conducted following the principles of the Declaration of Helsinki.

### 2.2. Evaluation of Factors Affecting Sugar Intake Behavior

Based on the theory of planned behavior [19], which suggests that behavioral intention is predicted to a large extent by subjective norms, attitude, and perceived behavioral control, and previous studies [14,16,17,26], a 5-point Likert-scored questionnaire form consisting of 12 factors and a total of 78 items measuring these factors was developed to evaluate the effects of these factors on sugar intake behavior. The definitions of these factors and the items are provided in Table 1.

### 2.3. Evaluation of Sociodemographic and Lifestyle Characteristics and Health Status

The demographic characteristics assessed included age, sex, education, occupation, marital status, and income. The lifestyle characteristics assessed were smoking, alcohol intake, and physical exercise. Additionally, the presence of any health condition that may affect eating and food choices, adherence to any diet, and food intolerances or allergies were assessed.

### 2.4. Sugar Intake Estimation

Usual sugar intake was estimated by asking about the intake of food groups identified as contributing most to free sugar intake. These food groups included biscuits, breakfast cereals, cakes, chocolate and bars, dumplings, dairy desserts, sweet spreads and sauces, yogurts, juices and smoothies, carbonated drinks, sweetened tea, coffee and herbal teas, jams, honey, molasses, marmalade, other beverages such as lemonade, and other desserts such as ashure and tahini halva. The following were the response options for all intake-related questions: “rarely or never”, “less than once a week”, “once a week”, “2–3 times a week”, “4–6 times a week”, “1–2 times a day”, “3–4 times a day”, and “more than 5 times a day”. The scores obtained from this questionnaire, which inquired about sugar-containing foods, were considered sugar intake.

### 2.5. Evaluation of Sugar-Sweetened Beverage (SSB) Intake

The beverage intake questionnaire (BEVQ-15) was used to evaluate the energy obtained from SSBs. This 15-item original questionnaire was developed and validated by Hedrik et al. [28]. In our study, the sugar-containing categories of the BEVQ-15, of which we have previously assessed the validity and reliability, were used. These categories included 100% fruit juices, sweetened juice beverages or drinks, soft drinks, diet soft drinks/artificially sweetened drinks, tea or coffee with cream or sugar, sweetened tea or coffee, and energy and sports drinks. Participants were asked about their frequency of SSB consumption in the past month and were asked to rate it on a 7-point Likert scale (0 = never and 6 = at least 3 times a day). Moreover, participants were asked to indicate the number of servings consumed. Total calories consumed from the questioned beverages were calculated using the consumption frequency (once per day) and the amount of beverage consumed (mL/time) for each beverage, and the mean daily score for the previous month was obtained.

### 2.6. Statistical Analysis

Data analysis was performed using the R version 4.3.1 program (R Core Team, Vienna, Austria) [29]. The normality of data was evaluated according to whether the skewness and kurtosis values were between −1 and +1. Confirmatory factor analysis (CFA) and the weighted least squares mean and variance method suitable for Likert-type ordinal data were employed, and the maximum likelihood robust method was used in structural equation modeling. The fit of the models was assessed using Chi-square with degrees of freedom ratio (χ^2^/df), root mean square error of approximation, root mean square of standardized residuals, goodness-of-fit index (GFI), adjusted GFI, comparative fit index, and the Tucker–Lewis index. Indirect effects of mediation analyses were calculated using the Sobel test. Johnson transformation was used for the normality transformation of the dependent variable. The fitness index and variance inflation factor were used for checking for multicollinearity, whereas the Durbin–Watson test was used for examining the presence of autocorrelation. The Spearman and Pearson correlations were used in correlational hypothesis testing. To estimate the effects, linear regression, and multiple robust regression using Huber’s M-estimator were employed. Interpretations were performed at a 95% confidence level.

**Table 1 foods-13-03109-t001:** Items and definitions of the variables evaluated in this study.

Variable	Items
Habit [30]	HA1: consuming foods and drinks high in free sugar as part of my daily diet is something I automatically perform. HA2: consuming foods and drinks high in free sugar as part of my daily diet is something I perform without having to consciously remember. HA3: consuming foods and drinks high in free sugar as part of my daily diet is something I perform without thinking. HA4: consuming foods and drinks high in free sugar as part of my daily diet is something I start to perform before I realize I am performing it.
Subjective norm [19]	SN1: most individuals who are significant to me would approve of me consuming foods and drinks high in free sugar as part of my daily diet. SN2: most individuals whose opinions I value believe that I should consume foods and drinks high in free sugar as part of my daily diet. SN3: most individuals who are significant to me are consuming foods and drinks high in free sugar as part of their daily diet.
Perceived behavioral control [19,31]	PBC1: it is mostly up to me whether I consume foods and drinks high in free sugar as part of my daily diet. PBC2: it would be possible for me to consume foods and drinks high in free sugar as part of my daily diet. PBC3: I have complete control over whether I consume foods and drinks high in free sugar as part of my daily diet. PBC4: if I wanted to, I could easily consume foods and drinks high in free sugar as part of my daily diet.
Intention [19]	IN1: I intend to consume foods and drinks high in free sugar as part of my daily diet in the next month. IN2: I expect to consume foods and drinks high in free sugar as part of my daily diet in the next month. IN3: it is likely that I will consume foods and drinks high in free sugar as part of my daily diet in the next month.
Self-control [32]	SC1: I have difficulty starting tasks. SC2: I immediately perform my chores. SC3: I find it difficult to get down to work. SC4: I am always prepared. SC5: I frequently waste my time. SC6: I start tasks right away. SC7: I tend to postpone decisions. SC8: I like to get to work at once. SC9: I need a push to get started. SC10: I tend to carry out my plans.
Personal impact [16]	PI1: I tend to crave sweet foods. PI2: I tend to crave sugars. PI3: I tend to crave sweeteners (removed). PI4: I want to reduce my sweet food intake. PI5: the presence or absence of sweet foods in my diet influences my mood. PI6: the presence or absence of sugars in my diet influences my mood. PI7: the presence or absence of sweeteners in my diet influences my mood (removed). PI8: I feel indifferent toward sweet foods. PI9: the sweet taste is physically addictive. PI10: sugar is physically addictive.
Personal management [16]	PM1: when I consume sugars, I balance out my diet through exercising and/or eating other healthy foods. PM2: when I consume sweeteners, I balance out my diet through exercising and/or eating other healthy foods. PM3: when I consume sweet foods, I balance out my diet through exercising and/or eating other healthy foods. PM4: my preference and/or intake of sugars depends on how much knowledge I have on them. PM5: my preference and/or intake of sweeteners depends on how much knowledge I have on them. PM6: I only consume sweet foods during special occasions. PM7: I only consume sugars during special occasions. PM8: I only consume sweeteners during special occasions. PM9: I categorize my sweet food intake into either “special” or “normal”. PM10: my health or body image will determine whether I modify my sugar intake or not. PM11: my health or body image will determine whether I modify my sweet food intake or not. PM12: my health or body image will determine whether I modify my sweetener intake or not. PM13: individuals who I am with (family, friends, and colleagues) influence my sweetener intake (removed).
Apathy [16]	AP1: individuals are highly concerned about cutting down on sweet foods. AP2: individuals are highly concerned about cutting down on sugars. AP3: individuals are highly concerned about cutting down on sweeteners. AP4: sugar is not as bad as fat for your health (removed). AP5: adding sugar in food products is unnecessary.
Negativity [16]	NE1: sweeteners are worse for your health than salt. NE2: sweeteners are physically addictive. NE3: sweeteners are not as bad as fat for your health (removed). NE4: adding sweeteners in food products is unnecessary. NE5: I feel guilty whenever I consume sweeteners. NE6: labels are misleading and deceptive. NE7: the food environment hinders me from reducing my sweetener intake.
Perceived understanding [14]	PU1: I know where to find credible information on sugars. PU2: I know where to find credible information on sweet foods. PU3: I know where to find credible information on sweeteners. PU4: If someone asks me, “what are sweeteners?”, I can explain. PU5: If someone asks me, “what is sugar?”, I can explain. PU6: I do not know whether to consume sugars or sweeteners (removed). PU7: I know how to replace sugars with sweeteners in cooking/baking. PU8: I know what strategies or policies have been implemented for reducing sugar intake in Turkiye.
Perceived nonautonomy [33]	PNA1: The desire or need for sweet food changes with age. PNA2: The desire or need for sugar changes with age. PNA3: The desire or need for sweeteners changes with age. PNA4: Completely eliminating sugar from my diet is impossible (removed). PNA5: Completely eliminating sweet food from my diet is impossible (removed).
Attitude [34]	ATT1: Consuming less sugary foods/drinks is a good thing for me.
	ATT2: Consuming less sugary foods/drinks is a healthy thing for me.
	ATT3: Consuming less sugary foods/drinks is something I enjoy.
	ATT4: Consuming less sugary foods/drinks is something I effortlessly perform.
	ATT5: Consuming less sugary foods/drinks is a delicious thing for me.
	ATT6: Consuming less sugary foods/drinks is something that is valuable to me.

Habit (HA): these are responses that individuals repetitively perform in their daily lives and that are quickly activated in memory compared with alternatives. Subjective norm (SN) is the social pressure perceived by an individual to perform or not to perform a behavior. Perceived behavioral control (PBC) is the perceived ease or difficulty of performing the behavior in relation to past experiences and expected barriers related to the behavior. Intention (IN) is an indicator of how much effort individuals are willing to exert or how much effort they plan to exert to perform a behavior. Self-control (SC) is the individual’s ability to create, manage, self-regulate, control, and govern within oneself to select the best option among alternatives. Personal impact (PI) is the internal or emotional orientation of an individual that is effective for performing and activating a behavior. Personal management (PM) refers to the management of factors in a manner that benefits the individual while performing a behavior. Apathy (AP) is defined as insufficient interest, resulting in individuals not responding to marketing messages. Negativity (NE) refers to negative judgments about performing a behavior. Perceived understanding (PU) refers to an individual’s actions of acquiring knowledge, comprehension, and application of a behavior. Perceived nonautonomy (PNA) refers to the inability of individuals to make and implement decisions on their own, independent of external influences. Attitude (ATT) is defined as an individual’s beliefs, feelings, and tendencies regarding an object or behavior.

## 3. Results

Overall, 942 questionnaires were submitted. Of these, two participants failed the logical statements. The participant characteristics for the remaining 940 questionnaires are described in Table 2. The majority of participants were women (women, 93.19%; men, 6.81%), with ages ranging from 18 to 72 (34.95 ± 9.89) years. The mean BMI of the participants was 24.82 ± 4.95 kg/m^2^, and 16.78% of them had food allergies.

Examining the items of the developed questionnaire according to reliability and convergent validity results revealed that an acceptable result was obtained with factor loadings of >0.3. Although PBC had a relatively low-performing reliability (CR, 0.638; α = 0.622), PBC, PI, PM, SC, and NE have relatively low-performance validities. Other variables presented high-performance validity and reliability values. Three items in PI (the seventh item appeared in CFA), one item in PM, one item in AP (apathy), one item in NE (Negativity), one item in PU, two items in PNA (perceived nonautonomy), and two items in ATT (Attitudes) were removed, which impaired high or low-performance validity and reliability. Figure 1 presents the results of the confirmatory factor analysis (CFA) for the constructs used in this study. Significant relationships between the variables in the model were found as a result of the CFA. After 42 modifications were made, the model fit was improved, ensuring that the model best reflected the theoretical structure. During the model improvement process, the ‘personal management’ construct and its related items were removed, along with the PI7 item (the seventh item in the personal impact construct) during the CFA process. The modifications were made to better clarify the relationships between the latent variables and the observed variables (Figure 1).

Examining the structural equation model (SEM) (Table 3 and Figure 2) showed that habits (β = −0.243; *p* < 0.001), subjective norms (β = −0.140; *p* < 0.001), perceived behavioral control (β = −0.138; *p* < 0.001), and perceived understanding (β = −1.270; *p* < 0.001) significantly negatively influenced attitude. Conversely, self-control (β = 0.033; *p* < 0.001), apathy (β = 0.560; *p* < 0.001), and negativity (β = 0.417; *p* < 0.001) positively affected attitude (*p* < 0.05). Additionally, attitude significantly negatively affected intention (β = −1.622; *p* < 0.001). Examining the effect of intention and attitude on sugar intake showed that, although intention significantly positively affected sugar intake (β = 1.489; *p* < 0.001), the effect of attitude on sugar intake was not significant (*p* = 0.666). The indirect negative effect of attitude on sugar intake through intention (β = −2.414; *p* < 0.001) was statistically significant.

Perceived understanding (β = −2.016; *p* < 0.001) significantly negatively influenced intention, whereas perceived nonautonomy (β = 1.709; *p* < 0.001), negativity (β = 0.658; *p* < 0.001), and apathy (β = 0.880; *p* < 0.001) showed a significant positive effect. Moreover, personal impact (β = 0.358; *p* < 0.001) significantly positively affected sugar intake.

As the direct effect of attitude on sugar intake was not significant in the SEM, but its indirect effect on intention was significant, the existence of the effect of attitude on sugar intake was investigated using the independent mediation model, and the full mediation relationship was interpreted (Table 4 and Figure 3). The null hypothesis regarding the direct effect of attitude on sugar intake, which was a prerequisite for mediation in the model, was rejected, and this effect was negative and significant (β = −0.209; *p* = 0.018). The effect of intention, which is another precondition, on sugar intake was significant and positive (β = 0.717; *p* < 0.001). The indirect effect of attitude on sugar intake through intention was negative and significant (ATT → IN → SI; β = −0.077; *p* = 0.002). The total effect comprising the direct effect of attitude and the indirect effect mediated by intention was negative and significant (X → M → Y + c’; β = −0.286; *p* = 0.001). This model explained 4.9% of the variance. Accordingly, although the existence of a direct significant effect of attitude alone on sugar intake, which was the mediation condition, was satisfied, the direct effect of attitude on sugar intake became not significant with the mediation of intention in the SEM. This finding was accepted as confirming that intention plays a full mediating role in the effect of attitude on sugar intake in the SEM.

The SEM diagram shows that habits, self-control, perceived behavioral control, subjective norms, apathy, negativity, and perceived autonomy significantly predict attitude, whereas attitude affects intention and sugar intake through intention. However, attitude did not significantly affect sugar intake directly (without intention). Accordingly, evaluating the structural equation diagram and the independent mediation model together revealed that attitude, influenced by various factors, reduced sugar intake through intention, and intention fully mediated this effect in the SEM (Figure 2).

The fit indices of the CFA model and SEM showed that the model was significant (*p* < 0.001). Based on the range of CFA fit indices, the model was confirmed in terms of fit indices with five Good Fits, one Perfect Fits, and one Marginally Acceptable Fits. Regarding SEM fit indexes, the model was confirmed, with three Good Fits and four Perfect Fits.

The relationships between consumers’ attitudes and preferences toward sugar intake and BMI, age, sugar intake, honey intake, low- or no-calorie-sweetener intake, total energy from beverages, and total scores are shown in Figure 3. A moderate positive relationship between sugar intake and habits (r = 0.336; *p* < 0.001); a weak positive relationship between sugar intake and subjective norms (r = 0.114; *p* < 0.001), intention (r = 0.278; *p* < 0.00), and personal impact (r = 0.196; *p* < 0.001); and a weak negative relationship between sugar intake and self-control (r = −0.216; *p* <0.001), perceived understanding (r = −0.082; *p* = 0.012), and attitude (r = −0.145; *p* < 0.001) were observed (Figure 4).

Analyzing the effect of demographic variables on sugar intake (Table 5) showed that when the average effect of all variables was zero, the average sugar intake was 29.733, and this value was a significant cutoff point. The effect of food allergy was significant in the multiple robust regression in Model 1 (*p* = 0.002), whereas other variables were not significant (*p* > 0.05). A simpler analysis of the effect of food allergy on sugar intake in Model 2 revealed a significant and decreasing beta coefficient (β = −0.303; *p* < 0.001). Based on the reference category, sugar consumption decreased, moving from respondents without food allergies to those with food allergies. This effect explains the very small portion of variance (1.3%).

## 4. Discussion

Here, we aimed to evaluate the effects of the theory of planned behavior [19] and variables examined in previous studies [14,16,17] on sugar intake in the population in Turkey. In this regard, we examined the effects of 10 variables, including habits, subjective norms, and perceived behavioral control, on attitude and intention toward sugar intake. As the items we used in the questionnaire were items that we prepared by considering the variables in previous studies [14,16,17] and whose validity had not been evaluated, we initially performed exploratory factor analysis and CFA. Subsequently, we continued the analysis by removing some items under some variables and the personal management variable from the model, as it did not fit the model structure.

The theory of planned behavior is based on the notion that attitude, subjective norms, and perceived behavioral control can predict the intention to perform a particular behavior [19]. In our study, we observed that subjective norms and perceived behavioral control significantly negatively affected attitude. The effect of attitude on intention was negative and significant. Examining the effect of intention on sugar intake revealed a positive and significant effect, whereas the effect of attitude on intake was not significant. Therefore, we evaluated using the independent mediation model and observed that attitude influenced sugar intake through intention. Similarly, a study by Phipps et al. [17], who examined the theory of planned behavior for predicting sugar intake in adolescents, reported that attitude, subjective norm, and perceived behavioral control had positive and significant effects on intention to intake sugar; however, intention was not associated with sugar intake. A study by Zoellner et al. [35], wherein they used the theory of planned behavior to explain the intake of SSBs, reported that 38% of the variance in beverage intake was explained by behavioral intention, perceived behavioral control, subjective norms, and attitude. These findings support the idea that the theory of planned behavior can predict sugar intake. The effect of attitude on intention being negative in our study is likely due to the reverse construction of the items related to attitude. Moreover, the reason why the effect of attitude on sugar intake is not significant in the SEM is that intention fully mediates this situation, as we showed in the mediation model. Intention alone has an increasing effect on sugar intake. However, when intention and attitude come together, intention acts as a full mediator and reduces sugar intake. This finding is because intention increases sugar intake, whereas attitude reduces this effect of intention. In other words, these findings indicate that intention and attitude should converge to reduce sugar intake.

Besides the variables in the theory of planned behavior, we included different variables in the model to complete this theory. In this regard, previous studies suggested that habits including family and home environment can be related to sugar intake [36,37]. Similarly, we determined that habits can significantly predict attitude; however, this interaction was negative. This negative aspect may again be due to the items being structured in the reverse direction. Another variable was personal impact, which refers to the internal and emotional factors influencing an individual to perform a behavior [16]. Regarding this issue, Tang et al. [16] reported in their study involving 581 adult participants that personal impact was negatively associated with reporting more frequent intake of sugary food groups. In the present study, wherein personal impact was assessed using sentences such as “I tend to crave sweet foods”, it was surprising that participation in this variable was lower in those who frequently consumed sugary foods. Our findings, unlike those of the study by Tang et al. [16], showed that personal impact has a significant positive effect on sugar intake; however, it does not have a significant effect on attitude. This finding suggests that personal impact can affect sugar intake without changing attitude.

Perceived understanding and autonomy, defined as an individual’s ability to make decisions independently of knowledge and understanding actions and external factors when performing a behavior, are other significant factors [14,33]. We showed that perceived understanding negatively affects attitude and intention. Regarding autonomy, previous studies indicated that this variable is associated with health-related behaviors, including fruit and vegetable intake, energy intake, physical activity, and smoking [38,39]. Additionally, our findings confirmed that perceived nonautonomicity affects attitude and intention, indicating that it can predict sugar intake. In addition to all these, positive and negative judgments toward a certain food are significant variables for consumer behavior [40]. Individuals’ value judgments and interest in healthy nutrition may lead to differences in their definitions of healthy nutrition and dietary patterns [41,42,43]. Accordingly, we assessed the effects of negativity and apathy on sugar intake behavior and showed that both factors positively affect attitude and intention. As the items measuring negativity and apathy are constructed in the same direction with attitude, it is an expected finding that they positively affect sugar intake behavior. However, it is a surprising finding that intention items positively affect sugar intake behavior despite being constructed in the opposite direction with these variables. Nevertheless, these findings indicate that individuals’ value judgments and interests affect their intake behaviors. These findings were supported by the negative or positive correlations of intention and sugar intake, with both the theory of planned behavior and other variables in the correlation analysis we performed, together using structural equation and mediation analyses. Evaluating all these data together revealed that the theory of planned behavior can predict individuals’ sugar intake behaviors. When the critical aspects of this theory in the literature [23,24] are considered and combined with different frameworks, sugar intake attitude and intention can be more clearly predicted with different aspects.

Sociodemographic factors are significant factors influencing consumers’ attitudes and intentions toward a product or action [44]. Park et al. used data from the National Health Interview Survey to assess added sugar intake-associated sociodemographic variables in adults in the US and reported that those who consumed more added sugar were younger, less educated, had lower income, were less physically active, and were current smokers [45]. Another study evaluating demographic factors and SSB intake showed that the intake rate of these beverages was higher in males, younger individuals, individuals with lower education levels, and smokers [46]. However, in our study, a significant effect of sociodemographic variables, including age, sex, education level, and smoking, on sugar intake was not observed. This finding is likely due to the fact that our population was mostly females, with education levels ranging from bachelor’s to postgraduate degrees, and non-smokers. Conversely, we determined that having a food allergy resulted in reduced sugar intake. Tang et al. evaluated attitudes toward sugar and sugar-sweetened foods and stated that having a health condition or food allergy did not affect sugar intake. However, it is also reported in the literature that having health problems, such as an allergy or intolerance, may affect consumer behavior [47]. Our findings also support this determination.

This study had some limitations. First, this study was conducted online, and all data were based on participants’ statements. Second, the population generally comprised women with higher levels of education and income. Therefore, this should not be ignored when generalizing the findings of this study to the population. Third, this study examined factors affecting individuals’ total sugar intake. It is also important to consider attitudes and intentions towards a particular food containing sugar.

## 5. Conclusions

We here aimed to predict individuals’ sugar intake behavior by combining the theory of planned behavior with other frameworks. We observed negative and significant effects of subjective norms and perceived behavioral control as defined in the theory of planned behavior on attitude. Moreover, examining the effect of intention on sugar intake showed a positive and significant effect, whereas attitude had no significant effect. Using mediation analysis, we noted that attitude affected sugar intake through intention. Furthermore, we determined that factors including habits, perceived understanding, perceived nonautonomy, negativity, and apathy can affect attitudes toward sugar intake. These findings suggest that the theory of planned behavior can be effective in predicting sugar intake behavior by supporting it with different frameworks. Moreover, although we could not show a significant effect of sex and education level, sociodemographic factors such as having food allergies may influence the sugar intake behaviors of individuals. However, when generalizing all these findings, it should be kept in mind that the population where this study was conducted represents a group with high levels of education and income.

## Figures and Tables

**Figure 1 foods-13-03109-f001:**
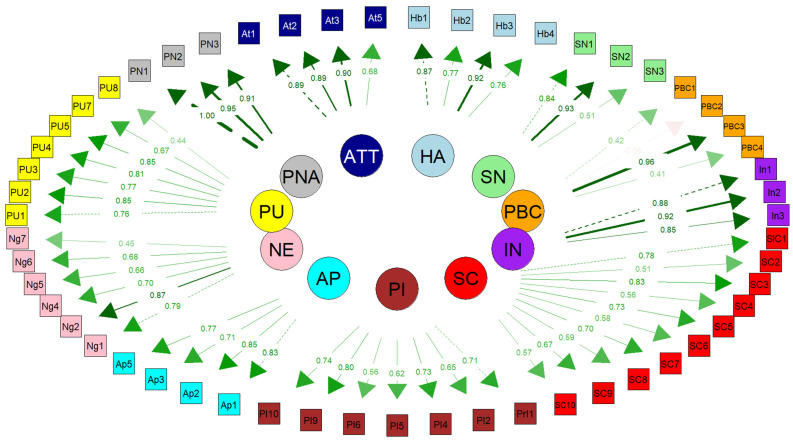
Confirmatory factor analysis questionnaire (By removing the PM variable and the 7th item in PI, the model was harmonized).

**Figure 2 foods-13-03109-f002:**
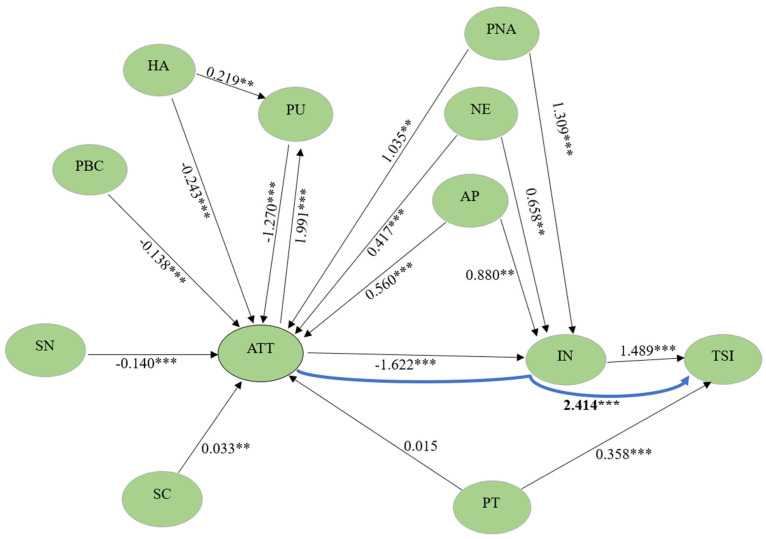
Structural equation modeling analysis effect diagram (Components such as habits, perceived understanding, perceived lack of autonomy, negativity, and apathy influenced attitude towards sugar intake). ** *p* < 0.01, *** *p* < 0.001.

**Figure 3 foods-13-03109-f003:**
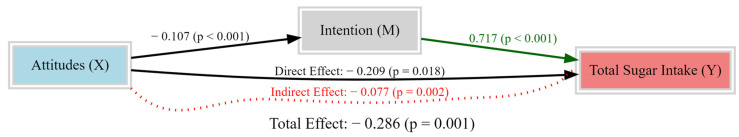
Independent mediation model for sugar intake (Mediation analysis showed that attitude influenced sugar intake through intention).

**Figure 4 foods-13-03109-f004:**
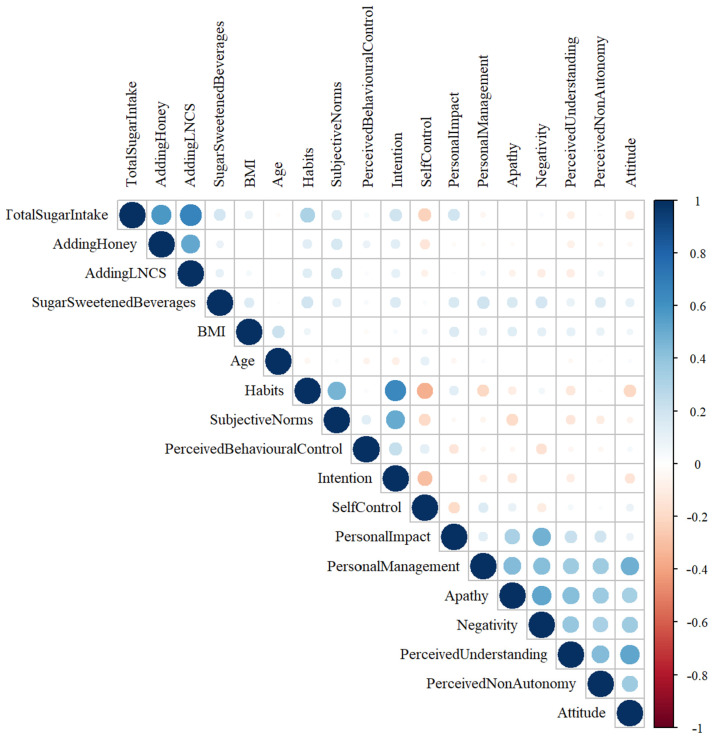
Relationship between sugar intake and various factors (Relationships were observed between sugar intake and habits, subjective norms, intention, personal impact, self-control, perceived understanding, and attitude).

**Table 2 foods-13-03109-t002:** Participants’ characteristics.

Characteristic	x^−^ ± SD or % (N)
**Sex**	
Woman	93.19 (876)
Man	6.81 (64)
**Age (years)**	34.95 ± 9.89
**Education level**	
Primary/secondary school	0.8 (7)
High school	7.42 (65)
Associate degree	2.28 (20)
Bachelor degree	59.25 (519)
MSc and Ph.D.	30.25 (265)
**Marital status**	
Single	42.81 (375)
Married	57.19 (503)
**Income**	
Student and pocket money	9.47 (83)
Below the minimum wage	8.22 (72)
TL 17,000–27,000	17.92 (157)
TL 27,001–37,000	8.33 (73)
TL 3700–47,000	10.39 (91)
TL > 47,000	45.66 (400)
**Smoking**	
No	74.2 (650)
Yes	25.8 (226)
**Regular alcohol intake**	
No	85.73 (751)
Yes	14.27 (125)
**Exercise 150 min a week**	
No	61.42 (538)
Yes	38.58 (338)
**Food allergy**	
No	83.22 (729)
Yes	16.78 (147)
**BMI**	24.82 ± 4.95
**Total sugar intake**	30.91 ± 12.87
**Adding honey**	1.42 ± 1.11
**Adding LNCS**	5.97 ± 3.39
**Sweetened beverage intake (kcal)**	98.16 ± 180.92

LNCS: low- or no-calorie sweetener.

**Table 3 foods-13-03109-t003:** Structural equation model.

Dependent ← Independent		β	SE	z-Value	*p*
**Baseline Model**	ATT ← HA	−0.243	0.064	−3.780	<0.001 ***
	ATT ← SN	−0.140	0.039	−3.551	<0.001 ***
	ATT ← PBC	−0.138	0.035	−3.881	<0.001 ***
	ATT ← SC	0.033	0.010	3.267	0.001 **
	ATT ← PI	0.015	0.009	1.663	0.096
	ATT ← AP	0.560	0.155	3.602	<0.001 ***
	ATT ← NE	0.417	0.088	4.739	<0.001 ***
	ATT ← PU	−1.270	0.299	−4.248	<0.001 ***
	ATT ← PNA	1.035	0.217	4.765	<0.001 ***
	IN ← ATT	−1.622	0.412	−3.935	<0.001 ***
	TSI ← IN	1.489	0.252	5.920	<0.001 ***
	TSI ← ATT	−0.071	0.165	−0.431	0.666
	TSI ← IN ← ATT	−2.414	0.737	−3.276	0.001 **
**Predictive Adjustments**	IN ← PU	−2.016	0.454	−4.446	<0.001 ***
	IN ← PNA	1.709	0.429	3.988	<0.001 ***
	IN ← NE	0.658	0.180	3.651	<0.001 ***
	IN ← AP	0.880	0.266	3.307	0.001 **
	PU ← ATT	1.991	0.147	13.509	<0.001 ***
	PU ← HA	0.219	0.076	2.868	0.004 **
	TSI ← PI	0.358	0.074	4.820	<0.001 ***

HA, habit; SN, subjective norm; PBC, perceived behavioral control; IN, intention; SC, self-control; PI, personal impact; AP, apathy; NE, negativity; PU, perceived understanding; PNA, perceived nonautonomy; ATT, attitude. ** *p* < 0.01, *** *p* < 0.001.

**Table 4 foods-13-03109-t004:** Independent mediation model for sugar intake.

Independent → Dependent	β	SE	t-Value	Std. β	*p*
X → M	−0.107	0.024	−4.395	−0.142	<0.001 ***
M → Y	0.717	0.118	6.102	0.196	<0.001 ***
X_(c’)_ →Y	−0.209	0.089	−2.360	−0.076	0.018 *
X → M → Y_a_	−0.077	0.025	−3.080	−0.028	0.002 **
X → M → Y + c’	−0.286	0.090	−3.199	−0.104	0.001 **

X, attitude; Y, sugar intake; M, intention; R^2^, 0.049. * *p* < 0.05, ** *p* < 0.01, *** *p* < 0.001.

**Table 5 foods-13-03109-t005:** Effects of participants’ characteristics on sugar intake.

**Model 1**	**β**	**SE**		**t**	**%95 Lower**	**%95 Upper**	** *p* **
(Intercept)	29.733	2.298		12.936	25.228	34.237	<0.001
Sex	1.958	1.385		1.414	−0.757	4.672	0.157
Age	−0.035	0.039		−0.904	−0.112	0.041	0.366
Education Level	0.229	0.431		0.531	−0.616	1.074	0.596
Marital Status	1.118	0.779		1.435	−0.409	2.645	0.151
Occupation	−0.148	0.272		−0.544	−0.680	0.385	0.587
Income	0.292	0.199		1.464	−0.099	0.682	0.143
Smoking	0.029	0.779		0.037	−1.497	1.555	0.970
Regular Alcohol Intake	−0.511	0.957		−0.534	−2.385	1.364	0.593
Exercise 150 min A Week	0.229	0.704		0.326	−1.151	1.610	0.745
Change In Dietary Habits In The Last 12 Months	−0.790	0.488		−1.618	−1.747	0.167	0.106
Health Problem	−1.011	0.905		−1.117	−2.784	0.762	0.264
Food Allergy	−2.888	0.934		−3.090	−4.719	−1.056	0.002 *
Currently Diet	1.052	0.753		1.397	−0.423	2.527	0.162
**Model 2**	**β**	**SE**	**Std. β**	**t**	**%95 Lower**	**%95 Upper**	** *p* **
(intercept)	0.052	0.036		1.463	−0.018	0.122	0.144
food allergy ref: no (0)	−0.303	0.086	−0.114	−3.525	−0.471	−0.134	<0.001 *

Model 1: Dependent variable: sugar intake, Multiple robust regression using Huber’s M-estimator, * *p* < 0.001. Model 2: Dependent variable: sugar intake (Johnson transformed), Linear regression; R^2^: 0.013; F: 12,426; model * *p* < 0.001.

## Data Availability

The original contributions presented in this study are included in the article/Appendix A, and further inquiries can be directed to the corresponding author.

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
