# Peer review of "Factors Driving Individuals’ Attitudes toward Sugar and Sweet-Tasting Foods: An Analysis within the Scope of Theory of Planned Behavior"

_foods, 2024, doi:10.3390/foods13193109_

Round 1

Reviewer 1 Report

Comments and Suggestions for Authors

This study need the following revisions:

You have to mention the main results (values) of your research in the abstract.

Was the employed survey validated for the study population? Did you perform any pre-tests? This has to be mentioned and clarified.

Figures 1, 2, 3, and 4 need to be explained and their quality has to be improved.

The applied survey should be included as Supplementary Material

Study strengths and limitations should be provided in the Discussion section. The authors speak superficially about the limitations, but they should go into more depth.

References should be formatted according to the journal’s guidelines as well as tables and the text (for example, the title should be properly formatted).

Author Response

Dear Reviewers,

We would like to thank you for the insightful comments and suggestions. We made all possible changes that were suggested and detailed the changes below. We believe that the manuscript is substantially improved after making the suggested edits.

We would like to thank you again for your valuable time and insight to strengthen our paper.

Sincerely,

The Authors

Comments and Suggestions for Authors

Reviewer: 1

  1. You have to mention the main results (values) of your research in the abstract.

Authors >> In line with your suggestion, we have added the values of the main findings to the abstract.

Was the employed survey validated for the study population? Did you perform any pre-tests? This has to be mentioned and clarified.

Authors >> For the questionnaire presented in Table 1, a validity and reliability studies were conducted in this population and the findings were described in lines 253-268 of the manuscript. Briefly, some factors (3 items in PI, 1 item in PM, 1 item in NE, 1 item in PU, 2 items in PNA, 2 items in ATT) were removed from the questionnaire in line with the analysis. However, in the confirmatory factor analysis, the PM factor and the 7th item in PI were removed. This finding is shown in figure 1.

Figures 1, 2, 3, and 4 need to be explained and their quality has to be improved.

Authors >> Explanations for figures were given in parentheses (to avoid confusion with the text). Also the quality of the figures has improved.

The applied survey should be included as Supplementary Material

Authors >> Survey added to Supplementary Material.

Study strengths and limitations should be provided in the Discussion section. The authors speak superficially about the limitations, but they should go into more depth.

Authors >> More specific additions have been made to the limitations section based on your suggestion.

References should be formatted according to the journal’s guidelines as well as tables and the text (for example, the title should be properly formatted).

Authors >> At your suggestion, we have realigned the tables. We also have revised the references and changed the writing style and layout.

Reviewer 2 Report

Comments and Suggestions for Authors

Authors conducted an interesting work on the factors driving individuals’ attitude toward sweet-tasting foods. The paper needs to be revised or adjusted as follows.

1. The table format of the entire text is inconsistent. Please adjust it. For Table 1, Definitions include too much text, please place them at the bottom of the table in the form of annotations. Items should provide key points rather than the entire sentence.

2. The clarity of figures is insufficient, which affects reading. For Figure 1, please represent the factors in different colors and prioritize them accordingly. For Figure 2, the direction of the arrow is too messy, please adjust it. For Figure 4, the text is too small to be clear, please adjust the font size.

3. In this study, what are the basic principles for sample selection? Will it affect the research results?

4. There are so many types of sugar and sweets, how can the author eliminate or reduce the influence of differences in food composition on the research results?

Comments on the Quality of English Language

Minor editing of English language required.

Author Response

Dear Reviewers,

We would like to thank you for the insightful comments and suggestions. We made all possible changes that were suggested and detailed the changes below. We believe that the manuscript is substantially improved after making the suggested edits.

We would like to thank you again for your valuable time and insight to strengthen our paper.

Sincerely,

The Authors

Comments and Suggestions for Authors

Reviewer: 1

  1. The table format of the entire text is inconsistent. Please adjust it. For Table 1, Definitions include too much text, please place them at the bottom of the table in the form of annotations. Items should provide key points rather than the entire sentence.

Authors >> We have realigned the tables. We also have placed the definitions of the factors at the bottom of the table.

  1. The clarity of figures is insufficient, which affects reading. For Figure 1, please represent the factors in different colors and prioritize them accordingly. For Figure 2, the direction of the arrow is too messy, please adjust it. For Figure 4, the text is too small to be clear, please adjust the font size.

Authors >> In line with your suggestion, the figures have been reorganized and their quality has been improved. Descriptions have also been added.

  1. In this study, what are the basic principles for sample selection? Will it affect the research results?

Authors >> The minimum sample size for this study was calculated as 390 for 78 items, 5 times the total number of items in the questionnaire. A total of 940 participants were reached through social media. Most of the participants were women, with high levels of education and income. It was stated in the limitations section that this situation should be taken into consideration in terms of the generalizability of the findings to the society.

  1. There are so many types of sugar and sweets, how can the author eliminate or reduce the influence of differences in food composition on the research results?

Authors >> We questioned the foods containing sugar that are commonly consumed in Turkey. From the scoring of this food frequency questionnaire, we made an estimate of total sugar intake. We also inquired about the frequency of consumption of sugar-containing beverages and calculated the total calorie intake from these beverages, which we believe provides data on overall sugar intake rather than on a specific food.

Round 2

Reviewer 1 Report

Comments and Suggestions for Authors

The manuscript has been improved. It can be published in Foods

Author Response

Dear Reviewer, 

Thank you for your support and contributions.

Reviewer 2 Report

Comments and Suggestions for Authors

Please conduct some chemical composition tests and provide the composition of the sugars or sweets used in this study.

Comments on the Quality of English Language

Minor editing of English language required.

Author Response

Dear Reviewer,

We would like to thank you for the insightful comments and suggestions. We made all possible changes that were suggested and detailed the changes below. We believe that the manuscript is substantially improved after making the suggested edits.

We would like to thank you again for your valuable time and insight to strengthen our paper.

Sincerely,

The Authors

Comments and Suggestions for Authors

Reviewer: 2

  1. Please conduct some chemical composition tests and provide the composition of the sugars or sweets used in this study.

Authors >> We estimated the total sugar intake of the participants by scoring them in the food consumption frequency questionnaire, which queried about sugar-containing foods. We selected the foods in this questionnaire from the sugar-containing food groups commonly consumed in Turkey according to the Turkey Nutrition and Health Survey (2017) data. In addition, these foods are included in the high-sugar food category as they contain more than 10 grams of sugar per 100 grams as defined in the Turkish Food Codex (2017). Therefore, we think that the foods we queried are high-sugar food groups. We also questioned the frequency of consumption of sugar-containing beverages with the beverage consumption survey and calculated the calories from these beverages. However, if you want additional analysis, we can add other statistical analyses.

  1. Minor editing of English language required.

Authors >> Dear reviewer, before sending our manuscript to Foods, we sent it to a company called Enago for language correction support. We are sending you the certificate regarding the subject.
